# Electromyographic Diaphragm and Electrocardiographic Signal Analysis for Weaning Outcome Classification in Mechanically Ventilated Patients

**DOI:** 10.3390/s25196000

**Published:** 2025-09-29

**Authors:** Alejandro Arboleda, Manuel Franco, Francisco Naranjo, Beatriz Fabiola Giraldo

**Affiliations:** 1Faculty of Engineering, Universidad Autónoma de Bucaramanga (UNAB), Bucaramanga 680003, Colombia; aarboleda@unab.edu.co (A.A.); mfranco20@unab.edu.co (M.F.); 2Department of Automatic Control (ESAII), Barcelona East School of Engineering (EEBE), Universitat Politècnica de Catalunya (UPC), 08019 Barcelona, Spain; 3Clínica FOSCAL, Floridablanca 681003, Colombia; fnaranjo888@unab.edu.co; 4Institute for Bioengineering of Catalonia (IBEC)—The Barcelona Institute of Science and Technology, 08028 Barcelona, Spain; 5CIBER de Bioingeniería, Biomateriales y Nanomedicina (CIBER-BBN), 28903 Madrid, Spain

**Keywords:** cardiorespiratory characterisation, time-varying spectral analysis, mechanical ventilation, weaning outcome, electromyography, diaphragm, signal analysis, classification, Naive Bayes classifier

## Abstract

Early prediction of weaning outcomes in mechanically ventilated patients has significant potential to influence the duration of treatment as well as associated morbidity and mortality. This study aimed to investigate the utility of signal analysis using electromyographic diaphragm (EMG) and electrocardiography (ECG) signals to classify the success or failure of weaning in mechanically ventilated patients. Electromyographic signals of 40 subjects were recorded using 5-channel surface electrodes placed around the diaphragm muscle, along with an ECG recording through a 3-lead Holter system during extubation. EMG and ECG signals were recorded from mechanically ventilated patients undergoing weaning trials. Linear and nonlinear signal analysis techniques were used to assess the interaction between diaphragm muscle activity and cardiac activity. Supervised machine learning algorithms were then used to classify the weaning outcomes. The study revealed clear differences in diaphragmatic and cardiac patterns between patients who succeeded and failed in the weaning trials. Successful weaning was characterised by a higher ECG-derived respiration amplitude, whereas failed weaning was characterised by an elevated EMG amplitude. Furthermore, successful weaning exhibited greater oscillations in diaphragmatic muscle activity. Spectral analysis and parameter extraction identified 320 parameters, of which 43 were significant predictors of weaning outcomes. Using seven of these parameters, the Naive Bayes classifier demonstrated high accuracy in classifying weaning outcomes. Surface electromyographic and electrocardiographic signal analyses can predict weaning outcomes in mechanically ventilated patients. This approach could facilitate the early identification of patients at risk of weaning failure, allowing for improved clinical management.

## 1. Introduction

Mechanical ventilation (MV) is an essential life-saving therapy that reduces the work of breathing, maintains cardiopulmonary interaction, and ensures adequate gas exchange through the complex process of patient-mechanical ventilator interaction. Most patients admitted to the intensive care unit often require MV as organ support due to respiratory failure, hemodynamic instability, or both [1]. MV is indicated for various conditions, including severe pneumonia, acute respiratory distress syndrome, exacerbation of chronic obstructive pulmonary disease, cardiogenic shock, and neuromuscular diseases [2]. In particular, the demand for invasive mechanical ventilation increased significantly during the COVID-19 pandemic [3]. The duration of mechanical ventilation support varies from short-term to long-term, depending on the patient’s underlying condition and response to treatment.

During MV, the diaphragm, the primary muscle of respiration, plays a crucial role in generating the necessary inspiratory effort to maintain adequate ventilation. However, challenges often arise during the weaning process, particularly at the stage of extubation, where successful liberation from the ventilator is contingent on the patient’s ability to sustain independent respiration. Current guidelines recommend a two-step strategy for ventilator weaning [1]. Initially, patients were evaluated to determine their readiness for weaning. Subsequently, a spontaneous breathing test (SBT) is performed as a diagnostic test to determine the probability of successful extubation [4]. In general, extubation readiness tests are intended to create a respiratory load sufficient to mimic the patient’s post-extubation condition. In adult and paediatric patients, the presence of positive end-expiratory pressure or pressure support during an extubation readiness test significantly underestimates the patient’s respiratory burden after extubation. The use of an SBT has been reported to have a sensitivity of 97% but a low specificity (40%) for predicting readiness for extubation. Therefore, the decision to extubate is often based on subjective clinical judgement, with variability in extubation practices [5]. Extubation failure, characterised by the need for reintubation after an unsuccessful extubation trial, has significant clinical implications and is associated with adverse outcomes [6,7].

Electromyographic signals, which represent the electrical activity of the diaphragm, have gained considerable attention in recent research. Analysing diaphragmatic activity allows the evaluation of respiratory muscle function and prediction of extubation success in intubated patients [8]. Surface electromyography (sEMG) has been proposed as a completely non-invasive alternative for monitoring the efforts of the inspiratory and expiratory muscles using electrodes placed on the skin surface [9,10]. sEMG is a non-invasive method that offers a means to assess muscle status. Therefore, it can be used as an early biomarker of diaphragm fatigue. The parameters most commonly used to determine muscle condition from EMG are the root mean square (RMS) and mean frequency [11,12]. Studies have shown that sEMG can effectively monitor diaphragmatic electrical activity with minimal applicability concerns for evaluating respiratory mechanical loading or unloading and respiratory sensation during MV [13]. Other studies focusing on cardiorespiratory measures and extubation outcomes in preterm infant populations observed significant differences in heart rate variability (HRV) parameters, specifically very low-frequency power and sample entropy, between infants who had successful extubation and those who required reintubation [14]. They also noted that the change in diaphragmatic activity during a spontaneous breathing test in prematurely born infants can predict subsequent extubation failure with moderate sensitivity and specificity [5]. These studies suggest that further research is needed to test the accuracy of sEMG as a clinical diagnostic technique for decision-making in MV liberation in adult populations. Future studies should examine the performance of sEMG in predicting MV outcomes, such as success and failure of MV liberation [9,10].

In our previous studies, we investigated the weaning process from mechanical ventilation by analysing diaphragmatic electromyographic signals and extracted parameters, incorporating a correlation analysis between different physiological signals to enhance our understanding of the process [15,16]. Additionally, respiratory flow databases containing electrocardiogram (ECG) and respiratory flow signals from mechanically ventilated patients were used [17,18,19,20]. We propose to analyse sEMG diaphragm and ECG signals for the classification of weaning outcomes in mechanically ventilated patients.

In addition to collecting physiological signals, such as sEMG and ECG data, signal processing involves a combination of linear and nonlinear analyses. Linear methods, including power spectral density (PSD) and mean square coherence (MSC), were employed alongside nonlinear techniques such as sample entropy (SampEn) and approximate entropy (ApEn). These analyses provide information on the complexity and dynamics of the recorded signals, facilitating subsequent classification tasks aimed at distinguishing between successful and unsuccessful weaning outcomes in mechanically ventilated patients.

The use of spectral and nonlinear parameters in the analysis of physiological signals has a clear clinical justification in the context of ventilator weaning. MSC allows for the evaluation of synchronisation between physiological signals, such as diaphragmatic activity and heart rate, which provides key information for anticipating successful extubation. In patients undergoing spontaneous breathing trials, statistically significant differences in MSC were found between sEMG and ECG-derived respiration (EDR) in groups with successful and failed extubation [15]. Another study involving 121 patients showed that MSC between HRV and respiratory flow, combined with principal component analysis (PCA), achieved 92% accuracy in classifying weaning outcomes [18]. Unlike univariate metrics, such as amplitude or mean frequency, MSC captures complex and functional physiological interactions between systems like the heart and diaphragm.

In contrast, entropy (ApEn and SampEn) measures the complexity and unpredictability of physiological signals, reflecting an organism’s adaptive capacity. In ventilator weaning, these parameters have been proposed as biomarkers for detecting respiratory muscle fatigue and autonomic dysfunction [21,22]. Their sensitivity to subtle changes surpasses that of linear metrics, enabling earlier detection of physiological instability.

Although the present study is based on the use of traditional classifiers such as Naive Bayes, Support Vector Machines (SVM), and K-Nearest Neighbours (KNN), selected for their interpretability and ability to identify physiologically relevant parameters, we acknowledge the growing potential of deep learning architectures as complementary tools for future research. In particular, non-invasive physiological signals, such as sEMG and ECG, have proven to be highly informative sources for neuromuscular and cardiorespiratory functions, with significant clinical value in predictive analyses. Their multichannel nature, high temporal resolution, and inherently sequential structure make them especially suitable for analysis using deep learning models such as Convolutional Neural Networks (CNN), recurrent networks like Long Short-Term Memory (LSTM), and, more recently, attention-based architectures such as Transformers. Several studies have reported promising results in predicting the outcomes of spontaneous breathing trials (SBT) using these architectures based on physiological signals, such as respiratory rate, oxygen saturation (SpO_2_), and heart rate [23,24]. In this regard, the incorporation of these approaches into multicentre studies could facilitate the development of more robust, generalisable, and clinically applicable predictive models.

## 2. Material and Methods

### 2.1. Protocol

To evaluate cardiorespiratory behaviour in mechanically ventilated patients for extubation, the following protocol was used. The proposed protocol aimed to simultaneously record three derivatives of electrocardiographic (ECG) signals and the superficial electromyographic (sEMG) signal of the diaphragm in patients who were undergoing mechanical ventilation. Signal acquisition was carried out with the patients in the supine position and a 35° head of bed elevation. To ensure complete recording of diaphragmatic activity, five equidistant surface electrodes were placed along the diaphragm muscle, with each channel corresponding to a specific electrode placement. The electrode configuration is shown in Figure 1.

Patients eligible for recording met the criteria for undergoing a spontaneous breathing test (SBT) as a preliminary step toward ventilator weaning, based on clinical judgement. All patients were on pressure support ventilation (PSV), a patient-triggered mode limited by pressure and cycled by flow. As the patient adapted to the support pressure, levels were gradually reduced in steps of 2 to 4 cmH_2_O until the patient reached minimal pressure support. Physiological recordings were continuously obtained when the patient was considered a candidate for maintaining respiratory function under end-pressure support. The ventilation settings included a positive end-expiratory pressure (PEEP) of 5 cmH_2_O, a fractional inspired oxygen concentration (FiO_2_) of 40%, flow trigger sensitivity of 2 L/min, and final pressure support of 8 cmH_2_O. The SBT protocol lasted an average of 60 min, and the final decision on extubation was made by the attending physician [15,16].


*Inclusion criteria:*


Patients were included if they had been on mechanical ventilation for at least 48 h and were considered ready for an SBT prior to weaning, according to the following clinical criteria:‑Resolution of the underlying cause of respiratory failure, without the need for vasopressors or sedatives (suspended 24 h before the study)‑Adequate oxygenation, PaO_2_ > 60 mmHg with an inspired oxygen fraction (FiO_2_) < 0.4 and a positive pressure at the end of the expiration (PEEP) < 8 cmH_2_O and PaO_2_/FiO_2_ > 150‑Cardiovascular stability: heart rate < 130 beats per minute and average blood pressure > 60 mm Hg‑Afebrile and hemodynamically stable‑Adequate haemoglobin level > 8 g/dL‑Adequate respiration muscle function‑Normal basic acid and electrolyte measurements


*Exclusion criteria:*


Patients were excluded if they met any of the following criteria:‑Age < 18 years, known pregnancy, protected adult‑Brain damage defined by a Glasgow Coma Scale  <  9‑Severe obesity‑Presence of a neuromuscular disease‑Suspected or confirmed phrenic nerve lesion‑Ongoing extracorporeal membrane oxygenation‑Contraindication for surface electrode placement

Patients who were unable to sustain spontaneous breathing were reconnected to mechanical ventilation, while those who successfully maintained spontaneous breathing were extubated. If a patient remained spontaneously breathing for 48 h after extubation, the weaning process was classified as successful; otherwise, reintubation was required, and the attempt was deemed unsuccessful.

### 2.2. Data Acquisition and Patients

A total of 40 patients participated in the study (25 males: 57.4 ± 20.6 years; 15 females: 67.9 ± 20.2 years). The study was carried out at the Intensive Care Medicine Department of Fundación Oftalmológica de Santander-FOSCAL, following a protocol approved by the local ethics committee. This study adhered to the principles of the Declaration of Helsinki for research involving human subjects, and informed consent was obtained from all participants.

Based on clinical criteria, the patients were classified into two groups: the successful group (SG; *n* = 19), who maintained spontaneous breathing for more than 48 h after extubation, and the failure group (FG; *n* = 21), who failed the SBT or required reintubation within 48 h after extubation. Table 1 presents the demographic characteristics of the patients classified according to their weaning outcomes. Tidal volume (VT) and respiratory rate (RR) are parameters provided by the mechanical ventilator and clinical data.

ECG signals were recorded at a sampling frequency of 128 Hz, and diaphragmatic sEMG signals were sampled at 1 kHz. Diaphragmatic sEMG signals were acquired using a wireless 5-channel surface electrode system (BTS FREEEMG 100 RT), and ECG recordings were obtained using a 3-lead Holter device EDAN SE-2003 (EDAN Instruments, Inc., Shenzhen, China). Figure 2 and Figure 3 show examples of recorded signals from a successfully extubated patient (ECG leads I, II, and III; sEMG from five equidistant channels).

Figure 2 presents an example of ECG signals recorded from patients with successful and failed extubation. Although both signals appear visually similar at first glance, differences in their regularity and stability are apparent. In the case of the patient with successful extubation, ECG leads I, II, and III exhibit a more consistent rhythmic pattern, with relatively uniform amplitudes and less variability between consecutive beats. Moreover, the morphology of the QRS complexes remains stable throughout the recording period, and the R–R intervals show reduced dispersion. In contrast, patients with failed extubation presented ECG signals with more pronounced variability. Although an apparent sinus rhythm is maintained, marked fluctuations in the amplitude of the QRS complexes are observed, particularly in lead II, with evident variations at approximately 5–10 s and again near 40 s of the recording. Additionally, subtle irregularities in the R–R intervals are detected. Since the visual differences may be subtle, the subsequent analysis focused on ECG-derived variables such as ECG-derived respiration (EDR) and heart rate variability (HRV).

Figure 3 shows significantly higher electromyographic amplitudes in the patient with failed extubation compared to the successful case, particularly in channels Ch1 and Ch2. This increase in EMG activity reflects elevated respiratory effort, which is characterised by the progressive and compensatory recruitment of respiratory motor units to maintain adequate ventilation. The observed high amplitude suggests early signs of muscle fatigue, indicating that the patient has reached the limit of their respiratory reserve [8].

### 2.3. Signal Analysis

Both the ECG and sEMG signals were subjected to a 60 Hz notch filter. For the ECG, a 0.05 Hz high-pass filter was applied to remove mains interference, while for the sEMG, a 20 Hz high-pass filter was used for the same purpose. The sEMG signals were resampled at 128 Hz to match the ECG sampling frequency. In addition, linear trending was eliminated, and internal preprocessing tools were used to reduce the noise, artefacts, and spikes. Finally, all signals were monitored and corrected as required.

Initially, 40 patients were recruited, and four were excluded due to technical problems with signal acquisition, leaving a final sample of 36 patients for analysis. Physiological signal processing involved extracting key electromyographic and cardiorespiratory parameters. ECG signals were used to obtain EDR and HRV. The EDR signal was computed from QRS complexes by measuring peak-to-peak amplitudes and applying piecewise cubic Hermite interpolation [25]. HRV was determined by interpolating the time series of the inverse RR interval at 1 Hz using a cubic spline function, with outliers exceeding 18% of the mean corrected recursively [26].

The sEMG signals were analysed to extract the electromyographic envelope (EMGe) and interpolated electromyographic signal (EMGi). EMGe was obtained by detecting peak amplitudes in the sEMG signal, followed by spline interpolation and quadratic polynomial smoothing using a 9 ms moving window [27]. EMGi was derived by identifying peaks above a set threshold with a minimum separation of 0.15 sto improve the continuity of the signal.

After processing the signals, the signal-to-noise ratio (SNR) was computed to evaluate the quality of the acquired data. The SNR, expressed in decibels (dB), is estimated as the ratio between the signal power and noise power, assuming that the noise is embedded in the signal. This calculation is based on the estimation of the total signal power and noise variance obtained by fitting a baseline model using the least-squares method [28]. The average SNR was calculated for EMG and ECG signals, as well as for derived signals including EMGe, EMGi, HRV, and EDR, in order to quantify the quality of each signal type throughout the study.

Respiratory activity was assessed using cross-correlation analysis. By computing the cross-correlation between EMGe signals from the five channels and EDR signals from the three ECG leads, it was determined that the sEMG signal from channel 3 and lead II of EDR best represented respiratory activity in mechanically ventilated patients.

### 2.4. Linear Analysis Methods

#### 2.4.1. Frequency Domain Methods Characterisation

Power spectral density (PSD) analysis was performed using the Welch method, which employs a modified periodogram with a Hanning window. PSD was estimated by averaging periodograms over 300-s segments with 50% overlap [15,16,27]. The key spectral parameters extracted were peak power (Hp), peak frequency (Fp), power within the modulation band (P), and up/down slopes (Us and Ds). The analysis focused on two main frequency bands: Low Frequency (LF: 0.04 Hz < f < 0.15 Hz) and High Frequency (HF: 0.15 Hz < f < 0.4 Hz), which are commonly used in physiological signal analysis (Figure 4).

Table 2 presents the spectral analysis parameters for EDR, HRV, EMGe, and EMGi, which characterise the modulation of respiratory, cardiac, and diaphragmatic activities during mechanical ventilation weaning.

#### 2.4.2. Coherence

Mean square coherence (MSC) was used to quantify the linear correlation between signals in the frequency domain. MSC was computed as(1)$$\;K_{xy}\left(f\right)\;=\;\frac{{{|S}_{xy}|}^{2}}{S_{xx}\left(f\right)S_{yy}\left(f\right)},$$
where $S_{xx}\;$and $S_{yy}$ represent the PSDs of the respective signals, and $S_{xy}$ is the cross-power spectral density [29]. The coherence between the EMGe, EMGi, EDR, and HRV signals was evaluated for the LF and HF bands to assess the interaction between cardiac and diaphragmatic functions (Table 3).

### 2.5. Nonlinear Analysis Methods

#### 2.5.1. Approximate Entropy

The approximate entropy (ApEn) was used to quantify the regularity and complexity of time-series data. It is calculated on a given time scale and depends on three key input parameters: N, representing the data length; m, which indicates the length of the vector window for diverse comparisons; and r, which denotes tolerance. ApEn was computed as:(2)$$\;ApEn\left(m,r,N\right)\;=\;-\;\frac{1}{N-m}\sum_{i\;=\;1}^{N-m}log\frac{A_{i}}{B_{i}}$$
where $B_{i}(r)$ represents the probability that two sequences remain similar for m points, and $A_{i}(r)$ extends this probability to m+1 points, both considering self-counting. Self-counting ensures that each segment is compared with all blocks in the sequence, including itself [29].

ApEn analysis was performed using a vector window of length m = 1 or 2 and a tolerance r set between 0.1 and 0.2 times the standard deviation of the signal, as recommended for physiological time-series data [26,30].

#### 2.5.2. Sample Entropy

Sample entropy (SampEn) was used to assess the complexity of time-series data by quantifying the unpredictability of fluctuations. SampEn is defined as the negative logarithm of the conditional probability that two similar sequences of m points remain identical at the next point (m+1), excluding self-matching [29]. SampEn was computed as:(3)$$\;SampEn\left(m,r,N\right)\;=\;\;-\log\frac{A^{m}(r)}{B^{m}(r)},$$
where $B^{m}(r)$ represents the probability that two sequences remain similar for *m* points, and $A^{m}(r)$ extends this probability to m+1 points. This ratio represents a conditional probability. Unlike ApEn, SampEn remains relatively consistent and is less dependent on data length. While developed to reduce biases present in ApEn, both measures exhibit conceptual similarities [31].

ApEn and SampEn analyses were performed using a window size of m = 2 and tolerance r = 0.15, following standard recommendations for physiological time-series analysis. Table 4 presents the parameters obtained from nonlinear analysis methods applied to EDR, HRV, EMGe, and EMGi, providing insights into the complexities of the cardiac and diaphragmatic signals.

### 2.6. Statistical Analysis

The statistical analysis included mean, coefficient of variation, kurtosis, and interquartile range to assess signal distribution and variability (Table 5). A total of 320 parameters were extracted for classification. Parameters with *p*-values < 0.05 were considered significant. Spearman’s rank correlation was applied, and only parameters with a correlation coefficient below 0.6 were selected for classification.

### 2.7. Classification Methods

The optimal subset of parameters was selected by evaluating different combinations using various classifiers, including Support Vector Machine (SVM) with a polynomial kernel, K-Nearest Neighbour (KNN), and Naive Bayes. For KNN, multiple distance metrics were considered, including city block, Chebyshev, correlation, cosine, Euclidean, Hamming, Jaccard, Mahalanobis, Minkowski, and Spearman [32]. A sequential selection approach was used to identify the most discriminative parameters that enhanced the accuracy of the model. Parameters were iteratively removed from the initial set until the refined subset achieved the highest possible classification performance. At each iteration, the parameter with the least impact on model performance was eliminated. This process allowed the identification of the most relevant parameters, resulting in improved accuracy, reduced data dimensionality, and simplified model complexity. The selected classifiers are discussed in the Section 2.7.1.

#### 2.7.1. SVM Classifier

Support Vector Machine (SVM) constructs hyperplanes to separate and classify patients into success or failure categories. This classification is achieved using a kernel function that maps the original data into a higher-dimensional space, facilitating the identification of an optimal separating hyperplane [33]. Several kernel functions were explored, including linear, Gaussian (radial basis function), and polynomial kernels. The choice of kernel was dataset-dependent, with different options tested to maximise classification performance. Among them, the polynomial kernel of degree 5 achieved the highest sensitivity values. The fundamental principle of SVM modelling is to determine the optimal hyperplane that maximises the margin between two classes in a binary classification problem [34]. Consider a training dataset comprising *n* observations, as follows:(8)$$\;\left\{{(x}_{1},y_{1}),\;\ldots \;,{(x}_{n},y_{n1})\right\},\;x_{i}\in R^{d},y_{i}\in \left\{-1,+1\right\},\;i\;=\;1,\;\ldots ,n$$
where $x_{i}$ represents the training observations, and $y_{i}$ denotes their corresponding labels.

The SVM model seeks the hyperplane decision given by the equation $\left(x\cdot w\right)+b\;=\;0$, which separates the two classes with the maximum margin according to $f\left(x\right)\;=\;\left(x\cdot w\right)+b$.(9)$$\;f\left(x\right)\;=\;\left\{\begin{array}{c} f\left(x\right)\geq 0,\;⟹x\in Class+1\\ f\left(x\right)<0,\;⟹x\in Class-1 \end{array}\right.$$
where w represents the normal vector of the hyperplane and b denotes the constant bias.

#### 2.7.2. KNN Classifier

KNN is an algorithm that operates on the principle of proximity, leveraging the distances between data points in multi-dimensional space. It identifies the closest instances or nearest neighbours from the original training dataset to make predictions or classifications. As the name suggests, KNN is a model that works locally based on the closest instances of all data objects in the original dataset used for training [35]. The kNN classifier is a non-parametric method [15,16]. Let $D^{n}\;=\;\left\{x_{1},\ldots ,x_{n}\right\}\;$ be a set of labelled prototypes (training samples). Also, let $x^{*}$ be the sample nearest to $\left(x\right)$. The objective is to assign × to the label associated with $x^{*}$. Specifically, for a set of labelled prototypes $\left\{\left(x_{1},y_{1}\right),\ldots ,\left(x_{n},y_{n}\right)\right\}$ where $\left(y\right)$ represents the label (class), then the kNN decision rule is given by:(10)$$\;k^{*}\;=\;argmin\left[dist\left(x,\;x_{i}\right)\right],$$
where *k** represents the optimal number of nearest neighbours and *dist*(·) is a distance metric. In the current work, *k** and the type of distance metric will be determined by using Spearman.

#### 2.7.3. Naive Bayes Classifier

The Naive Bayes classifier is a significant supervised learning technique based on Bayes’ theorem, which is named after Thomas Bayes. It operates under the assumption that the parameters are statistically independent [35]. As a probabilistic method, it assumes that the presence of a specific parameter in a class is independent of the presence of other parameters. This classifier can probabilistically predict the class of an unknown sample by leveraging the available training sample set to calculate the most probable output [36]. The Naive Bayes classifier can probabilistically predict the class of an unknown sample using the available training sample set to calculate the most probable output [36]. The most probable class $\left(c\right)\;$of an unknown sample with the conjunction $a_{1},\;a_{2},\ldots ,\;a_{m}$ is determined by:(11) CNB = argmax p(c|a1, a2, …, am)c ∈C      

Figure 5 presents a schematic overview of the methodology, outlining its key phases.

## 3. Results

This study analysed sEMG diaphragm and ECG signals to classify weaning results in mechanically ventilated patients based on clinical criteria. The results present linear and nonlinear analyses, identifying distinct patterns in respiratory, muscle, and cardiac activity between the successful and failed weaning groups.

### 3.1. Signal Analysis Results

The average SNR value for the processed EMG signals was 13.56 ± 4.22 dB, whereas that for the ECG signals was 21.21 ± 3.74 dB, indicating a higher signal-to-noise ratio in the cardiac signals. The SNR was calculated as the ratio between the signal power and noise power following the preprocessing of the signals. For the EMG-derived signals, an average SNR of 14.72 ± 4.32 dB was obtained for EMGe and 13.71 ± 2.55 dB for EMGi. Regarding the ECG-derived signals, the values were 18.39 ± 3.12 dB for HRV and 21.79 ± 2.57 dB for EDR. These results fall within the acceptable ranges reported in the literature, where an SNR ≥ 10 dB is considered sufficient for reliable EMG signal analysis [37,38] and an SNR ≥ 20 dB is desirable to ensure good quality in ECG signal processing [39,40].

The EDR was extracted from the ECG signal to characterise respiratory activity and compare it with electromyographic signals. The analysis revealed that the patients in the successful group exhibited a higher EDR amplitude [−17.3–38.4] mV), suggesting a reduced need for mechanical ventilation compared to the failure group, which presented a significantly lower amplitude range [−3.6–2.4] mV).

Regarding electromyographic activity, the EMGe amplitude was notably higher in the failure group [−2.2–2.4 × 10^−5^] mV) than in the success group [−1.7–1.9 × 10^−5^]. Additionally, a greater oscillation in diaphragmatic muscle activity was observed in the success group [−1.3–1.2 × 10^−5^] mV, indicating a higher frequency of this activity compared to the failure group [3.7–5.6 × 10^−5^] mV.

Cross-correlation analysis between EDR and EMG signals allowed the identification of significant relationships between cardiopulmonary activity and diaphragmatic muscle activity. Cross-correlation values exceeding 0.4 were found between EDR lead II and EMG signal interpolation channel 3 [15]. EDR lead II provides information on the structures of the heart’s septal wall, while EMG channel 3, located at the xiphisternal joint, captures muscle activity in the diaphragm attachment region.

Figure 6 illustrates the resultant EMGe and EMGi signals from channel 3, along with HRV and EDR from lead II obtained by coherence analysis. These signals capture the electromyographic and cardiorespiratory dynamics of mechanically ventilated patients.

In the group of patients who failed, a greater amplitude [5.9 × 10^−5^] mV was observed in diaphragmatic muscle activity, indicating a potentially greater difficulty in the extubation process [1.2 × 10^−5^] mV for the successful groups. In contrast, a higher oscillation was detected in the diaphragmatic muscle activity in the successful group, implying a higher frequency of diaphragmatic contractions.

### 3.2. Statistical and Correlation Analysis of Parameters Extracted from Linear and Nonlinear Methods

Figure 7 illustrates the EMGe, EMGi, HRV, and EDR signals along with their respective PSD for illustrative purposes, featuring one successful extubation case and one unsuccessful case. PSD of the successful group exhibits greater modulation in the high-frequency band, indicating an increase in diaphragmatic muscle activity and, consequently, a greater capacity for spontaneous breathing without ventilatory support. Consistent with this, no modulatory effect is observed in the low-frequency band for the EMGi and EMGe signals. However, for EDR PSD, modulation is observed in both bands, with a stronger effect in the high-frequency band.

The PSD analysis revealed subtle but statistically significant differences between patients with successful and unsuccessful extubation. As shown in Figure 8, the PSD curves of both groups exhibit similar overall patterns, with the most notable differences concentrated in specific frequency bands rather than uniformly distributed across the spectrum.

In the HF band, the successful extubation group showed greater spectral modulation than the unsuccessful group. For the EMGi signal, a power amplitude of 0.55 was observed in the successful group versus 0.47 in the unsuccessful group, representing a 17% difference. The EMGe signal showed values of 0.44 and 0.39, respectively (13% difference), while the HRV signal presented values of 0.38 and 0.35 (8.5% difference). Although these differences may appear small in absolute terms and are difficult to distinguish visually, they represent physiologically meaningful variations.

The analysis of the LF band did not reveal significant modulatory differences in the EMGi and EMGe signals between the groups. However, the EDR signal exhibited distinctive behaviour, showing significant modulation in both frequency bands. The greater separation between EDR curves, particularly evident in the 0.2–0.4 Hz range, indicates that ECG-derived respiratory patterns capture more pronounced differences between groups, with a more marked effect in the HF band.

Figure 9 shows the average coherence curves for the group of patients who experienced one successful and one failed extubation attempt. It is evident that coherence is greater in successful cases than in unsuccessful ones. The MSC results show a coherence greater than 0.7, suggesting a close relationship between these signals during the extubation process. A stronger correlation is observed between EMGi amplitude and HRV and EDR measured parasympathetic activity in both groups of patients, with a higher correlation with HRV. Furthermore, the consistency between EMGe, EDR, and HRV is stronger in patients who achieved extubation than in those who failed.

A total of 320 parameters were extracted, including linear and nonlinear methods. Only those that demonstrated substantial importance were selected for further analysis. Four different types of statistical parameters were extracted from the EMGe, EMGi, EDR, and HRV signals. Here, of the 320 parameters, only 43 were selected as significant and correlated. Table 6 and Table 7 present the mean values, coefficient of variation, kurtosis, and interquartile range, highlighting statistically significant differences. These statistical parameters help distinguish between successful and failed classes. Differences were evaluated by comparing the mean values between the successful and failure groups (Table 6).

The power measure of the EMGi signals, both in the HF and LF frequency bands, was significantly higher in the successful group than in the failure group. For EMGe signals, the power in the LF frequency band behaved similarly, with a greater magnitude for the successful group.

The highest intensity spectral power for both EMGi, EMGe, and HRV signals in the LF frequency band was significantly higher in patients who were successfully extubated than in those who failed. The EMGi signals in the HF frequency band were also significantly higher in the successful group. A different pattern was observed in HRV signals in the HF frequency band, where its density was significantly higher in patients who failed extubation.

The coherence between the EMGi and EDR signals, the EMGe and HRV signals, and the EMGe and EDR signals was significantly higher in patients who successfully completed the mechanical ventilation weaning process. In contrast, the coherence between EMGi and HRV increased in the group of patients who failed the weaning process. Higher coherence was observed in the EMGe, EMGi, HRV, and EDR signals in successful patients than in unsuccessful patients. This increased coherence suggests a stronger relationship between these signals during the extubation process in successful patients (Table 7).

Cardiac (${SampEn}^{HRV}$, ${ApEn}^{HRV}$) and diaphragmatic (${SampEn}^{EMGe}$, ${ApEn}^{EMGe},{\;SampEn}^{EMGi}$) complexities were significantly lower in patients who were unsuccessful in weaning compared to those who were successful, as observed in Table 8.

The Spearman rank correlation test was used to evaluate the correlations between the 43 parameters that were significant in the statistical testing (Table 9). Parameters with a correlation coefficient less than 0.60 were considered to have weak correlations and were consequently selected for the classifier performance evaluation. Of these, 15 parameters exhibited weak correlations.

### 3.3. Classifier Performance Evaluation

Three classifiers were used on two different datasets: one comprising patients who were successfully extubated and another comprising patients who failed extubation. Using 15 parameters with low correlation, a sequential selection process was carried out to identify optimal parameter combinations by entering them into three classifiers that increased the accuracy of the model. The parameters were iteratively removed until the subset achieved the highest performance. Results are detailed in Table 10, Table 11 and Table 12, with the Naive Bayes classifier demonstrating notably superior performance compared to the other two classifiers. The confusion matrix in Figure 10 illustrates the effect of classifiers on binary classification for the best result.

The classification performances of KNN, SVM, and a Naive Bayes classifier on the evaluation data are summarised in Figure 11. The results are based on 1000 runs for each classifier. In particular, excluding the input parameters from the classifier improved the performance of all classifiers. The Naive Bayes classifier achieved the highest classification performance after excluding eight parameters. In contrast, the KNN classifier achieves its highest performance by excluding six parameters, while the SVM classifier achieves its highest performance by excluding five parameters. However, these performances are lower than those of the Naive Bayes classifier. This analysis highlights the importance of parameters No. 10-41-17-29-32-33-39 for achieving a good classification performance with the Naive Bayesian classifier.

The parameters mentioned, such as the mean $P_{HF}^{EMGi}$, ${SampEn}^{EMGe}$, kurtosis ${Hp}_{HF}^{EDR}$, and the interquartile ranges of ${Fp}_{LF}^{EMGi}$, ${Hpcoh}_{HVR}^{EMGe}$, ${Hpcoh}_{EDR}^{EMGi}$, ${Hpcoh}_{HRV}^{EMGi}$ (0.4 Hz), contribute significantly to classifying weaning outcomes in patients with mechanical ventilation. These parameters were associated with greater success in mechanical ventilation weaning when there is increased diaphragmatic muscle power, greater complexity, and irregularity in the electromyographic signal of the diaphragm, reduced data dispersion, and increased coherence between the signals.

Subsequently, the seven parameters that exhibited better performance in the Naive Bayes classifier were utilised and tested in the KNN and SVM classifiers. The results presented in Table 13 indicate a superior performance for the Naive Bayes classifier.

Based on the results, the Naive Bayes classifier assisted in selecting the best uncorrelated parameters, which served as inputs for the SVM classifier. The SVM classifier demonstrated improved performance for these input parameters (see Figure 12). This study had certain limitations that warrant consideration. Despite efforts to ensure robustness, the sample size was relatively small, although it was consistent with that of comparable physiological investigations. Based on the parameters employed in the classifiers, the Naive Bayes classifier yielded the best results for identifying the most effective uncorrelated parameters. These findings underscore the potential utility of assessing cardiorespiratory measures prior to extubation, providing information on the prediction of extubation outcomes and informing clinical decision-making in patients undergoing mechanical ventilation.

## 4. Discussion

The findings of this work were consistent with those of previous work on the use of diaphragmatic electromyography during spontaneous breathing trials. During an SBT, the use of diaphragmatic electromyography has been studied to predict extubation failure in premature infants. The findings indicated that an increase in EMG amplitude during the SBT was associated with greater respiratory effort, suggesting that these infants might require more ventilatory support after extubation [5]. Accessory and expiratory muscle activation during SBT has also been examined, with findings indicating that increased electrical activity of the diaphragm was correlated with increased respiratory effort during the spontaneous breathing trial, highlighting its association with weaning failure [41].

Other authors, such as Dres M. et al. (2012) and Barwing J. et al. (2013), have delved into the intricacies of weaning failure, utilising diaphragmatic electrical activity (EAdi) as a metric to predict the outcome of weaning trials [8,42]. Their findings suggest that in the failure group, EAdi levels are higher from the onset of SBT, indicating an elevated initial respiratory demand. However, EAdi does not exhibit significant changes during the trial in either group. The second paper postulates that alterations in EAdi can predict SBT failure earlier than conventional parameters, highlighting that continuous EAdi monitoring can offer invaluable insights during weaning, aiding in the identification of patients who are not yet ready for discontinuation of respiratory support [42]. In line with this, Liu L. et al. (2012) examined neuroventilatory efficiency and readiness for extubation in critically ill patients, finding that patients who failed in extubation exhibited higher diaphragmatic electrical activity [43].

Furthermore, HRV was analysed, and it was found that patients in the successful group [5.4–6.5 mV] exhibited higher signal amplitude than those in the failed group [3.8–4.3 mV]. This finding is corroborated by Huang C. et al. (2014), who evaluated the association between HRV change and weaning outcomes in critically ill patients [44]. They found that HRV analysis in patients who failed the SBT showed a significant decrease in total power. A reduction in HRV variability was significantly associated with failure in the spontaneous breathing trial (SBT).

Da Silva R. et al. (2023), recently published a review to assess the usefulness of HRV as a predictor of MV weaning outcomes [45]. They found that patients who were successful in the weaning process exhibited better autonomic control of the heart, whereas those who experienced weaning failure showed an increase in the low-frequency components (LF) and a decrease in the high-frequency components (HF) of HRV.

Moreover, this study revealed an increased coherence between EMGi and HRV, as well as between EMGe and EDR, in patients who successfully underwent mechanical ventilation weaning compared to those who failed, suggesting greater interaction and synchronisation between diaphragmatic muscle activity and cardiovascular activity in patients who successfully underwent mechanical ventilation weaning. This result is consistent with previous findings indicating that the involvement of the diaphragm in cardiac function is often overlooked. It significantly influences HRV and other cardiovascular functions. In patients with heart failure, there appears to be reduced synchronisation between diaphragmatic and cardiac activity compared to healthy individuals [46].

Furthermore, the results showed that both cardiac and diaphragmatic complexities were markedly lower in patients who were unsuccessful in weaning than in those who were successful. Lower values of approximate entropy (ApEn) observed in the unsuccessful weaning group indicate higher regularity and predictability in the time-series data, implying a more ordered system. In contrast, higher ApEn values signify lower regularity and increased complexity in the data, indicating a less predictable system [47]. Lower ApEn values have been widely observed to be associated with the presence of pathologies, suggesting that under diseased or altered conditions, physiological systems tend to exhibit reduced entropy parameters [48]. Consistent with previous research, our findings align with the general trend that entropy parameters are significantly reduced in unhealthy patients compared to those with better health statuses [49,50]. A lower HRV entropy suggests an indicator of autonomic dysfunction in these patients due to the reduced variability in the intervals between heartbeats caused by mechanical ventilation. The decrease in diaphragmatic entropy may indicate a decrease in the adaptive capacity of the diaphragm and increased muscle fatigue in mechanically ventilated patients or those in the weaning process, which may have implications for the patient’s ability to breathe autonomously and affect their ability to tolerate extubation.

## 5. Conclusions

The results suggest that the analysis of time-varying signals, such as diaphragmatic electromyographic signals, can provide valuable information to predict success or failure in the extubation process in patients undergoing mechanical ventilation. The findings indicate differences in diaphragmatic muscle activity, modulation of cardiovascular signals, the relationship between signals, and the complexity of both systems in the successful and failed extubation groups. These results may have important clinical implications for improving the assessment and management of patients during mechanical ventilation weaning. In general, the study concluded that the analysis of diaphragmatic electrical activity and the regularity of time series can provide useful information for predicting extubation failure and difficult weaning. This study represents one of the initial efforts to use surface electromyography to monitor muscle activity in adult patients receiving assisted mechanical ventilation. Although our findings are promising, further research is imperative to enhance the fidelity of surface EMG signals and determine their practical value in predicting extubation outcomes.

This study has certain limitations that should be considered in the design of future research. First, the small sample size restricts the generalisation of the results to larger populations. Additionally, clinical factors that could influence weaning outcomes, such as comorbidities and specific respiratory parameters, were not included. For future studies, it is recommended to increase the sample size and perform a comprehensive analysis of the signals obtained from all five channels to assess their relevance in predicting extubation success or failure.

Another limitation of this study is the absence of external validation using independent datasets or comparisons with other clinical predictions. Although the results obtained show promising classification performance based on physiological signal features, we recommend that future studies incorporate independent cross-validation cohorts and establish direct comparisons with medical decision-making to improve the robustness and generalisation across diverse patient populations and clinical settings.

We emphasise the need for additional validation in larger multicentre cohorts before considering the implementation of these findings in intensive care units. While the results suggest potential clinical utility, they should be interpreted in conjunction with expert clinical judgement and within the specific context of each patient. The integration of algorithmic predictions with clinicians’ experience and real-time assessment remains critical to ensure safe and effective decision-making in the weaning process.

## Figures and Tables

**Figure 1 sensors-25-06000-f001:**
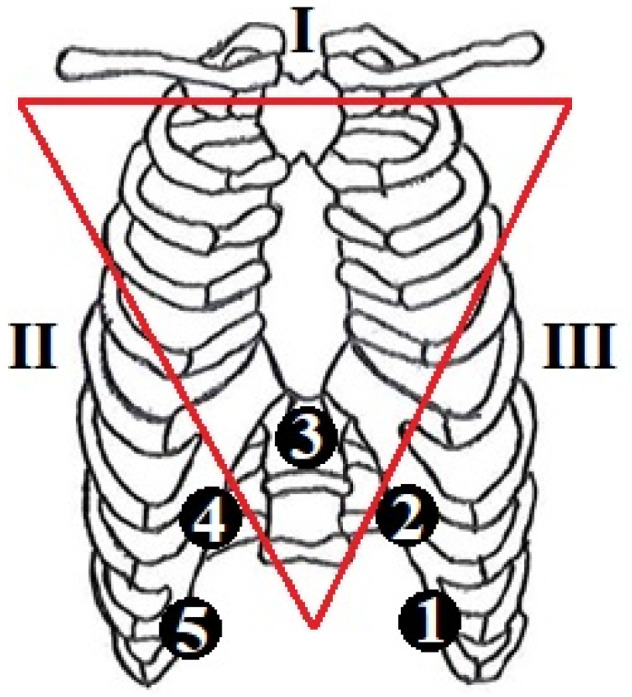
Acquisition scheme of ECG (leads I, II, and III) and diaphragmatic sEMG signals (equidistant channels 1 to 5).

**Figure 2 sensors-25-06000-f002:**
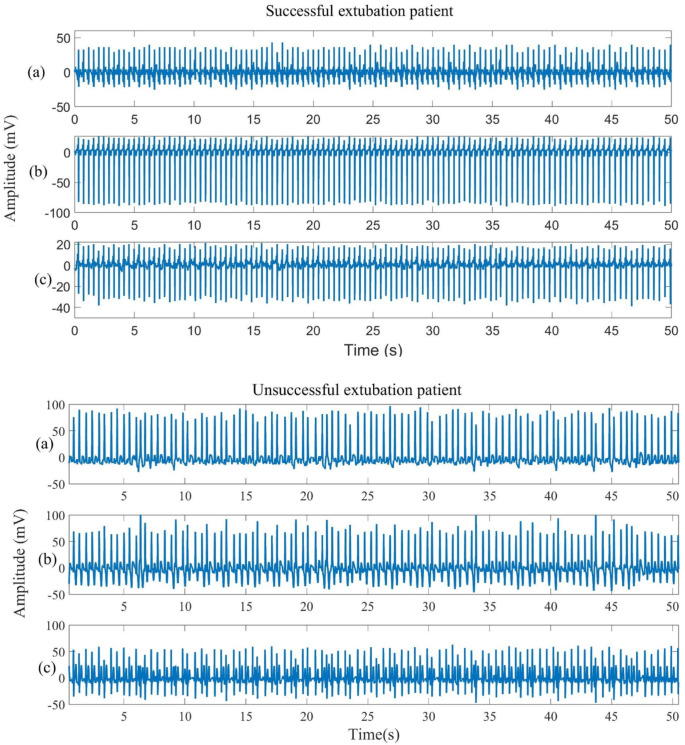
Sample results of signals from a successful and unsuccessful patient, after preprocessing, were obtained using ECG leads (**a**) Lead I, (**b**) Lead II, and (**c**) Lead III.

**Figure 3 sensors-25-06000-f003:**
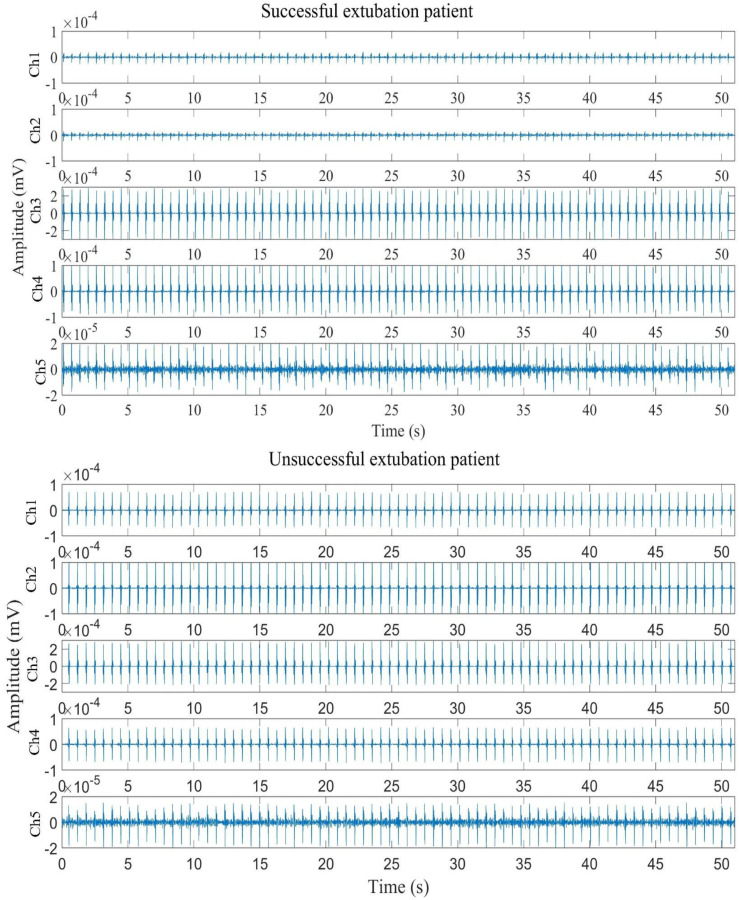
Sample results of signals from a successful and unsuccessful patient, after preprocessing, were obtained using five channels (Ch) of EMG around the diaphragm muscle.

**Figure 4 sensors-25-06000-f004:**
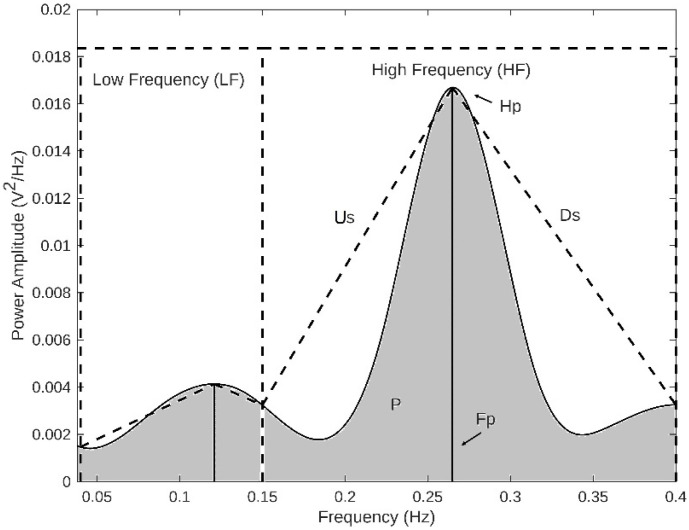
Parameters derived from spectral analysis in the high-frequency band. Fp represents the modulation frequency peak, P denotes the power within the modulation band, Ds corresponds to the down slope, and Us represents the up-slope.

**Figure 5 sensors-25-06000-f005:**
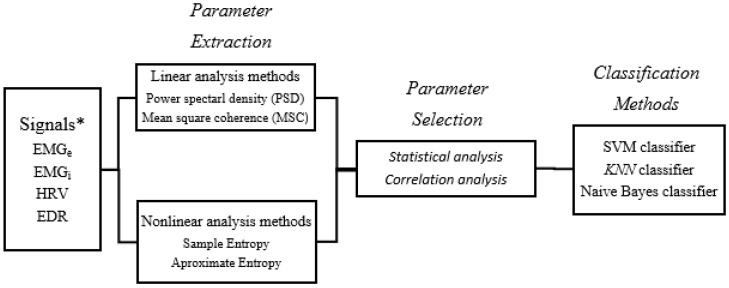
Main stages of parameter selection through principal component analysis, sequential selection of floating parameters, and classification process. * Surface diaphragm electromyographic signal envelope (EMGe), interpolated surface diaphragm electromyographic signal (EMGi), heart rate variability (HRV), and ECG-derived respiration (EDR).

**Figure 6 sensors-25-06000-f006:**
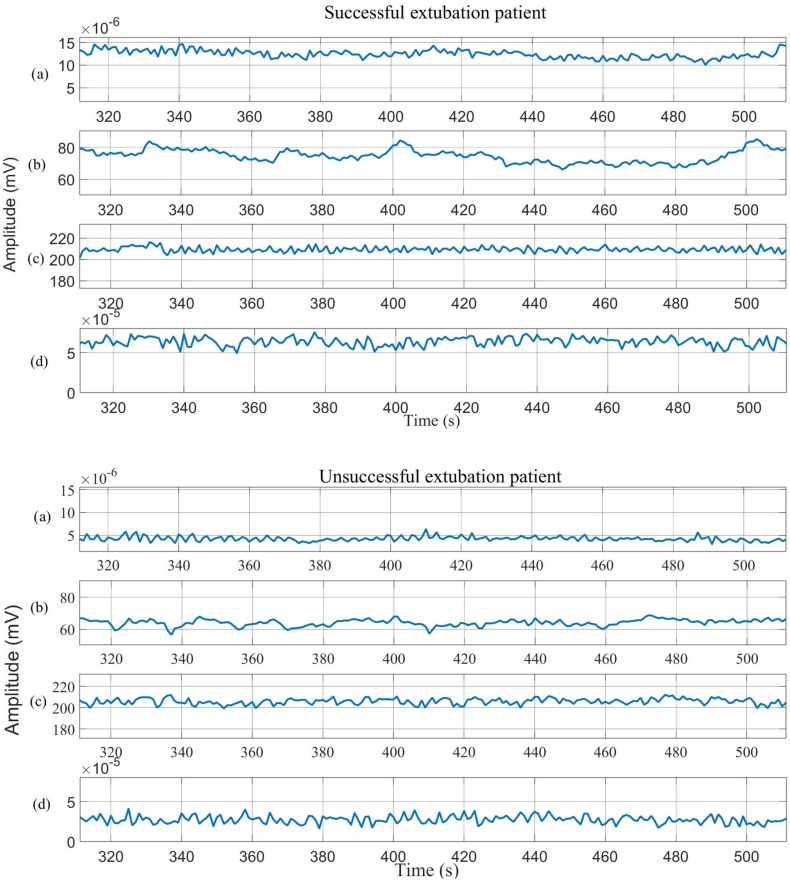
Illustrates the behaviour of (**a**) surface diaphragm electromyographic signal envelope (EMGe), (**b**) interpolated surface diaphragm electromyographic signal (EMGi), (**c**) heart rate variability (HRV), and (**d**) ECG-derived respiration (EDR) in successful and unsuccessful patients.

**Figure 7 sensors-25-06000-f007:**
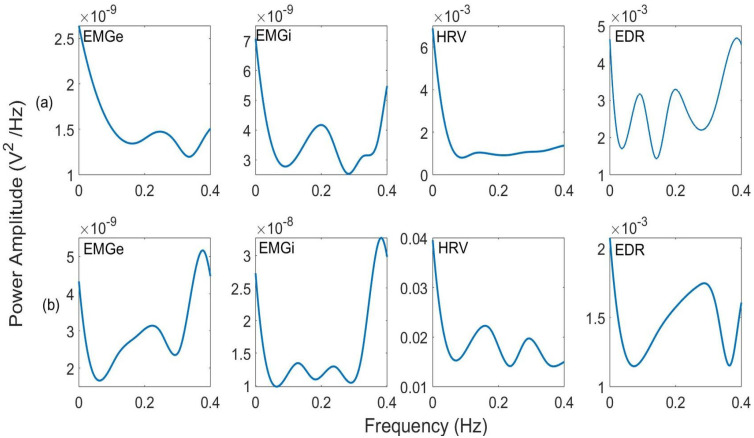
Illustrates the power spectral density (PSD) of the EMGe, EMGi, HRV, and EDR signals, providing an illustrative example of (**a**) a successful patient and (**b**) an unsuccessful patient.

**Figure 8 sensors-25-06000-f008:**
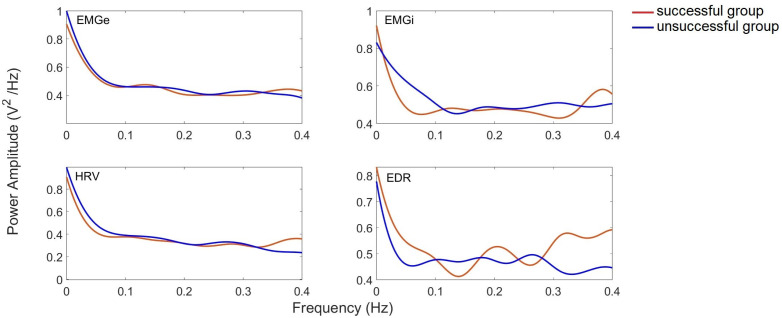
PSD amplitude averages for patients who experienced successful and unsuccessful extubation attempts.

**Figure 9 sensors-25-06000-f009:**
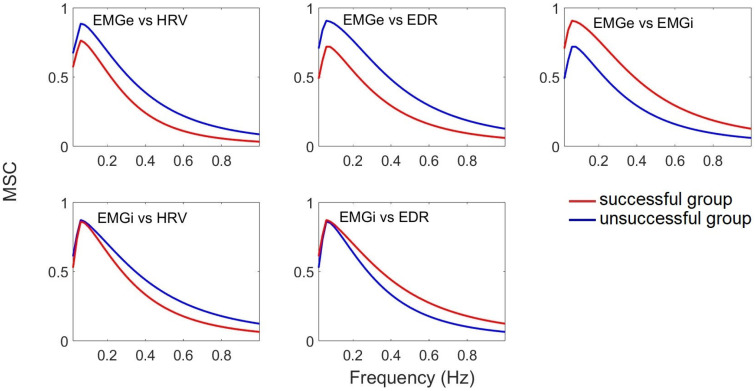
Coherence averages for patients who experienced successful and unsuccessful extubation attempts.

**Figure 10 sensors-25-06000-f010:**
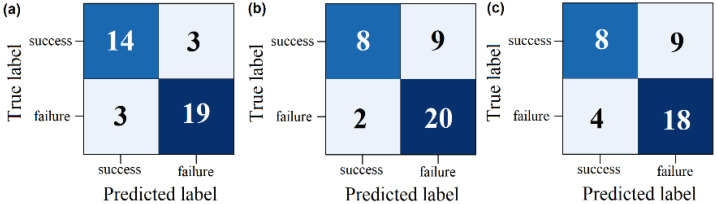
Confusion matrix for binary classification (successful group vs. failure group) for the following types of classifiers: (**a**) Naive Bayes, (**b**) KNN, and (**c**) SVM.

**Figure 11 sensors-25-06000-f011:**
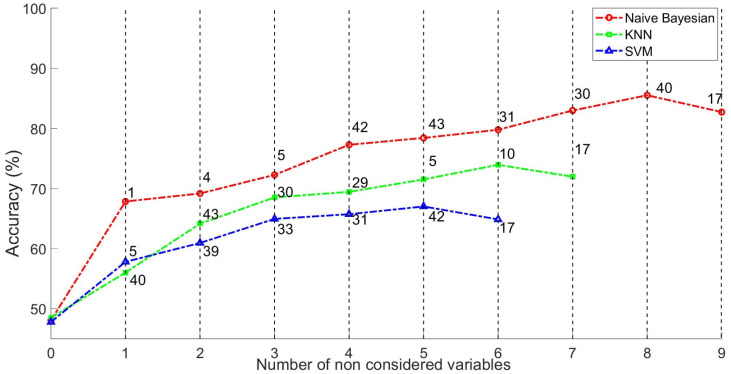
Performance comparison of classifiers (Naive Bayes, KNN, and SVM) for successful vs. failure group classification. In each iteration, the parameter to be excluded is shown in each curve of the classifier.

**Figure 12 sensors-25-06000-f012:**
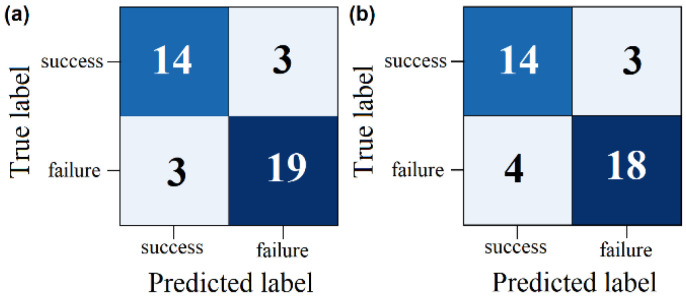
Confusion matrix for binary classification (successful group vs. failure group) using the best parameters obtained with the Naive classifier, No. 10-41-17-29-32-33-39, applied to the other two classifier methods: (**a**) KNN and (**b**) SVM.

**Table 1 sensors-25-06000-t001:** Demographic summary with mean ± SD values.

Class	Sex (M: Male; F: Female)	Age (Years) (Mean ± SD)	V_T_ (mL)	RR (rpm)
Successful group	12 M, 7 F	53.7 ± 23.3	539 ± 151.0	18.7 ± 3.2
Failure Group	13 M, 8 F	68.6 ± 15.7	498.3 ± 100.5	21.2 ± 3.6

V**_T_** = tidal volume (mL: millilitres) information provided by the mechanical ventilator, RR = respiratory rate (rpm: respiration per minute), SD = Standard Deviation.

**Table 2 sensors-25-06000-t002:** Parameters derived from spectral analysis.

Signal Analysis	Parameter	Definition
ECG-derived respiratory (EDR)	${Hp}_{LF}^{EDR}$	Highest peak modulation band at LF
	$P_{LF}^{EDR}$	Power of modulation band at LF
	${Fp}_{LF}^{EDR}$	Modulation frequency peak at LF
	${Us}_{LF}^{EDR}$	Slope between modulation frequency peak and start of the modulation frequency band at LF
	${Ds}_{LF}^{EDR}$	Slope between modulation frequency peak and end of the modulation frequency band at LF
	${Hp}_{HF}^{EDR}$	Highest peak modulation band at HF
	$P_{HF}^{EDR}$	Power of modulation band at HF
	${Fp}_{HF}^{EDR}$	Modulation frequency peak at HF
	${Us}_{HF}^{EDR}$	Slope between modulation frequency peak and start of the modulation frequency band at HF
	${Ds}_{HF}^{EDR}$	Slope between modulation frequency peak and end of the modulation frequency band at HF
Heart rate variability (HRV)	${Hp}_{LF}^{HRV}$	Highest peak modulation band at LF
	$P_{LF}^{HRV}$	Cardiac power at LF
	${Fp}_{LF}^{HRV}$	Modulation frequency peak at LF
	${Us}_{LF}^{HRV}$	Slope between modulation frequency peak and start of the modulation frequency band at LF
	${MDs}_{LF}^{HRV}$	Slope between modulation frequency peak and end of the modulation frequency band at LF
	${Hp}_{HF}^{HRV}$	Highest peak modulation band at HF
	$P_{HF}^{HRV}$	Cardiac power in HF
	${Fp}_{HF}^{HRV}$	Modulation frequency peak at HF
	${Us}_{HF}^{HRV}$	Slope between modulation frequency peak and start of the modulation frequency band at HF
	${Ds}_{HF}^{HRV}$	Slope between modulation frequency peak and end of the modulation frequency band at HF
EMG enveloped (EMGe)	${Hp}_{LF}^{EMGe}$	Highest peak modulation band at LF
	$P_{LF}^{EMGe}$	Power of modulation band at LF
	${Fp}_{LF}^{EMGe}$	Modulation frequency peak at LF
	${Us}_{LF}^{EMGe}$	Slope between modulation frequency peak and start of the modulation frequency band at LF
	${Ds}_{LF}^{EMGe}$	Slope between modulation frequency peak and end of the modulation frequency band at LF
	${Hp}_{HF}^{EMGe}$	Highest peak modulation band at HF
	$P_{HF}^{EMGe}$	Power of modulation band at HF
	${Fp}_{HF}^{EMGe}$	Modulation frequency peak at HF
	${Us}_{HF}^{EMGe}$	Slope between modulation frequency peak and start of the modulation frequency band at HF
	${Ds}_{HF}^{EMGe}$	Slope between modulation frequency peak and end of the modulation frequency band at HF
EMG Interpolated (EMGi)	${Hp}_{LF}^{EMGi}$	The highest peak modulation band at LF
	$P_{LF}^{EMGi}$	Power of the modulation band at LF
	${Fp}_{LF}^{EMGi}$	Modulation frequency peak at LF
	${Us}_{LF}^{EMGi}$	Slope between the modulation frequency peak and the start of the modulation frequency band at LF
	${Ds}_{LF}^{EMGi}$	Slope between the modulation frequency peak and the end of the modulation frequency band at LF
	${Hp}_{HF}^{EMGi}$	Highest peak modulation band at HF
	$P_{HF}^{EMGi}$	Power of the modulation band at HF
	${Fp}_{HF}^{EMGi}$	Modulation frequency peak at HF
	${Us}_{HF}^{EMGi}$	Slope between the modulation frequency peak and the start of the modulation frequency band at HF
	${Ds}_{HF}^{EMGi}$	Slope between the modulation frequency peak and the end of the modulation frequency band at HF

LF = low frequency range (0.04–0.15 Hz); HF = high frequency range (0.15–0.4 Hz).

**Table 3 sensors-25-06000-t003:** Parameters derived from the coherence analysis.

Analysis	Parameters	Definition
Cardiac and diaphragmatic interaction	${Pcoh}_{EDR}^{EMGe}$	Power of coherence between EMGe and EDR
	${Pcoh}_{HRV}^{EMGe}$	Power of coherence between EMGe and HRV
	${Pcoh}_{EDR}^{EMGi}$	Power of coherence between EMGi and EDR
	${Pcoh}_{HRV}^{EMGi}$	Power of coherence between EMGi and HRV
	${RMScoh}_{EDR}^{EMGe}$	Root mean square of the coherence between EMGe and EDR
	${RMScoh}_{HRV}^{EMGe}$	Root mean square of the coherence between EMGe and HRV
	${RMScoh}_{EDR}^{EMGi}$	Root mean square of the coherence between EMGi and EDR
	${RMScoh}_{HRV}^{EMGi}$	Root mean square of the coherence between EMGi and HRV
	${Fpcoh}_{EDR}^{EMGe}$	Modulation frequency peak of the coherence between EMGe and EDR
	${Fpcoh}_{HRV}^{EMGe}$	Modulation frequency peak of the coherence between EMGe and HVR
	${Fpcoh}_{EDR}^{EMGi}$	Modulation frequency peak of the coherence between EMGi and EDR
	${Fpcoh}_{HRV}^{EMGi}$	Modulation frequency peak of the coherence between EMGi and HVR
	${Hpcoh}_{EDR}^{EMGe}$	Highest peak modulation band of the coherence between EMGe and EDR
	${Hpcoh}_{HRV}^{EMGe}$	Highest peak modulation band of the coherence between EMGe and HRV
	${Hpcoh}_{EDR}^{EMGi}$	Highest peak modulation band of the coherence between EMGi and EDR
	${Hpcoh}_{HRV}^{EMGi}$	Highest peak modulation band of the coherence between EMGi and HRV
	${Uscoh}_{EDR}^{EMGe}$	Slope between modulation frequency peak and start of the modulation frequency band of the coherence between EMGe and EDR
	${Uscoh}_{HRV}^{EMGe}$	Slope between the modulation frequency peak and the start of the modulation frequency band of the coherence between EMGe and HRV
	${Uscoh}_{EDR}^{EMGi}$	Slope between modulation frequency peak and start of the modulation frequency band of the coherence between EMGi and EDR
	${Uscoh}_{HRV}^{EMGi}$	Slope between the modulation frequency peak and the start of the modulation frequency band of the coherence between EMGi and HRV
	${Dscoh}_{EDR}^{EMGe}$	Slope between modulation frequency peak and end of the modulation frequency band of coherence between EMGe and EDR
	${Dscoh}_{HRV}^{EMGe}$	Slope between the modulation frequency peak and the end of the modulation frequency band of the coherence between EMGe and HRV
	${Dscoh}_{EDR}^{EMGi}$	Slope between modulation frequency peak and end of the modulation frequency band of coherence between EMGi and EDR
	${Dscoh}_{HRV}^{EMGi}$	Slope between the modulation frequency peak and the end of the modulation frequency band of the coherence between EMGi and HRV
	${Hpcoh}_{EDR}^{EMGe}$(j)	Highest peak modulation band of coherence between EMGe and EDR.
	${Hpcoh}_{HRV}^{EMGe}$(j)	Highest peak modulation band of coherence between EMGe and HRV.
	${Hpcoh}_{EDR}^{EMGi}$(j)	Highest peak modulation band of coherence between EMGi and EDR.
	${Hpcoh}_{HRV}^{EMGi}$(j)	Highest peak modulation band of coherence between EMGi and HRV.

*j* corresponded to frequencies of 0.02 and 0.04 Hz.

**Table 4 sensors-25-06000-t004:** Parameters derived from nonlinear analysis methods.

Analysis	Index	Definition
Cardiac and diaphragmatic complexity	${SampEn}^{EDR}$	Sample entropy applied to ECG-derived respiration
	${ApEn}^{EDR}$	Approximate entropy applied to ECG-derived respiration
	${SampEn}^{HR}$	Sample entropy applied to heart rate
	${ApEn}^{HR}$	Approximate entropy applied to heart rate
	${SampEn}^{EMGe}$	Sample entropy applied to EMG enveloped
	${ApEn}^{EMGe}$	Approximate entropy applied to EMG enveloped
	${SampEn}^{EMGi}$	Sample entropy applied to EMG Interpolated
	${ApEn}^{EMGi}$	Approximate entropy applied to EMG Interpolated

**Table 5 sensors-25-06000-t005:** A brief description of the extracted parameters.

Feature	Detail	Formula	Equation No.
Mean	On average, the signal, it just adds all the samples in the signal and divides by the total number of samples n. In the discrete set of samples, the central values are the key values.	$\bar{x}\;=\;\frac{1}{n}\sum_{i\;=\;1}^{n}x_{i}$	(4)
Coefficient of variation	Normalised measures of distribution of data and defined as the ratio of standard deviation to the mean.	$CV\;=\;\frac{s}{\bar{x}}$	(5)
Kurtosis	It defines the peaks of the data distribution in our data. If the value of K is higher means, the peak is very sharp. We get the smooth curve of the data point if the value of K is less.	$k\;=\;\frac{\sum_{i\;=\;1}^{n}{\left(x_{i}-\bar{x}\right)}^{4}}{{ns}^{4}}$	(6)
Interquartile range	Measure of dispersion based on the lower and upper quartile. Distance between the 75th and 25th percentile in the sample	$IQR\;=\;Q_{3}-Q_{1}$	(7)

n is the number of samples on each trial. $\bar{x}\;$ represent means of signal. s  is the standard deviation of the signal. $Q_{1}$ is the first quartile of a distribution. $Q_{3}$ is the third quartile of a distribution.

**Table 6 sensors-25-06000-t006:** Time-varying signal processing results for mean values.

No.	Parameter	Successful Group (Mean [IQR])	Failure Group (Mean [IQR])	*p*-Value
1	${Hp}_{LF}^{HRV}$	50.61 [29.07–72.16]	50.36 [24.47–76.24]	0.0474
2	${Fp}_{HF}^{HRV}$	2.19 × 10^−1^ [1.62–2.77] × 10^−1^	2.77 × 10^−1^ [2.08–3.48] × 10^−1^	0.0078
3	${Hp}_{LF}^{EMGe}$	9.78 × 10^−9^ [−0.12–2.08] × 10^−8^	3.91 × 10^−9^ [0.28–7.54] × 10^−9^	0.0487
4	$P_{LF}^{EMGe}$	8.90 × 10^−7^ [−0.07–1.86] × 10^−8^	3.68 × 10^−7^ [1.32–6.05] × 10^−7^	0.0020
5	${Ds}_{LF}^{EMGe}$	−3.00 [−4.00–−2.00]	−2.99 [−4.82–−1.16]	0.0288
6	$P_{HF}^{EMGe}$	1.48 × 10^−6^ [0.09–2.86] × 10^−6^	7.12 × 10^−7^ [0.24–1.18] × 10^−6^	0.0069
7	${Hp}_{LF}^{EMGi}$	1.89 × 10^−8^ [−0.01–3.81] × 10^−8^	9.17 × 10^−9^ [0.38–1.44] × 10^−8^	0.0069
8	$P_{LF}^{EMGi}$	1.71 × 10^−6^ [0.15–3.27] × 10^−6^	9.24 × 10^−7^ [0.46–1.39] × 10^−6^	0.0106
9	${Hp}_{HF}^{EMGi}$	1.38 × 10^−8^ [0.30–2.47] × 10^−8^	9.54 × 10^−9^ [0.41–1.49] × 10^−8^	0.0356
10	$P_{HF}^{EMGi}$	3.01 × 10^−6^ [0.81–5.21] × 10^−6^	2.08 × 10^−6^ [1.09–3.07] × 10^−6^	0.0094
11	${Pcoh}_{EDR}^{EMGe}$	0.86 [8.04–9.30] × 10^−1^	0.73 [5.57–9.20] × 10^−1^	0.0003
12	${Pcoh}_{HRV}^{EMGe}$	0.88 [8.26–9.42] × 10^−1^	0.72 [5.26–9.16] × 10^−1^	0.0034
13	${Pcoh}_{EDR}^{EMGi}$	0.81 [7.56–8.82] × 10^−1^	0.17 [−0.07–0.35] × 10^−2^	0.0373
14	${Pcoh}_{HRV}^{EMGi}$	0.84 [7.32–9.53] × 10^−1^	0.87 [8.16–9.36] × 10^−1^	0.0470

**Table 7 sensors-25-06000-t007:** Time-varying signal processing results for coefficient of variation (CV), kurtosis (K), and interquartile range (IQR).

No.	Feature	Parameter	Successful Group (Mean ± SD)	Failure Group (Mean ± SD)	*p*-Value
15	CV	${Hpcoh}_{HRV}^{EMGe}$(0.4 Hz)	1.54 ± 0.0400	1.29 ± 0.0300	0.0021
16		${Hp}_{HF}^{EMGe}$	1.90 ± 0.0200	1.39 ± 0.0200	<0.0001
17	K	${Hp}_{HF}^{EDR}$	2.03 ± 0.0100	2.15 ± 0.0200	0.0003
18		${Hpcoh}_{EDR}^{EMGe}$	4.93 ± 0.0910	4.75 ± 0.1020	0.0199
19		${Hpcoh}_{EDR}^{EMGe}$(0.4 Hz)	3.01 ± 0.0830	3.85 ± 0.0980	<0.0001
20		${Hpcoh}_{HRV}^{EMGe}$(0.4 Hz)	1.87 ± 0.0030	2.49 ± 0.0050	<0.0001
21		${Dscoh}_{HRV}^{EMGe}$	1.98 ± 0.0090	3.12 ± 0.0150	<0.0001
22		${Hpcoh}_{EMGi}^{EMGe}$(0.02 Hz)	2.21 ± 0.0300	2.50 ± 0.0400	0.0015
23		${MHpcoh}_{EDR}^{EMGi}$(0.02 Hz)	2.43 ± 0.0300	2.61 ± 0.0200	0.0088
24		${MHpcoh}_{EDR}^{EMGi}$(0.4 Hz)	2.74 ± 0.0400	3.75 ± 0.0300	<0.0001
25		${MHpcoh}_{HRV}^{EMGi}$	4.93 ± 0.0500	4.40 ± 0.0600	<0.0001
26		${MHpcoh}_{HRV}^{EMGi}$(0.02 Hz)	1.98 ± 0.0080	2.52 ± 0.0090	<0.0001
27		${MHpcoh}_{HRV}^{EMGi}$(0.4 Hz)	1.56 ± 0.0090	2.54 ± 0.0180	<0.0001
28		${MDscoh}_{HRV}^{EMGi}$	1.62 ± 0.0080	2.89 ± 0.0060	<0.0001
29	IQR	${Fp}_{LF}^{EMGi}$	0.03 ± 0.0016	0.10 ± 0.0024	<0.0001
30		${Fp}_{HF}^{EMGi}$	0.12 ± 0.0023	0.20 ± 0.0034	0.0204
31		${Fp}_{HF}^{HR}$	0.13 ± 0.0029	0.24 ± 0.0035	0.0040
32		${Hpcoh}_{HVR}^{EMGe}$	0.09 ± 0.0006	0.26 ± 0.0008	0.0405
33		${Hpcoh}_{EDR}^{EMGi}$	0.00 ± 0.0003	0.02 ± 0.0003	0.0109
34		${Hpcoh}_{HRV}^{EMGe}$(0.02 Hz)	0.27 ± 0.0050	0.47 ± 0.0070	0.0170
35		${Pcoh}_{EMGi}^{EMGe}$	14.90 ± 0.1870	7.26 ± 0.2430	0.0361
36		${Hpcoh}_{EMGi}^{EMGe}$	0.38 ± 0.0100	0.19 ± 0.0600	0.0153
37		${RMScoh}_{HRV}^{EMGi}$	0.20 ± 0.0041	0.11 ± 0.0024	0.0182
38		${Pcoh}_{HRV}^{EMGi}$	21.7 ± 2.1500	10.8 ± 2.2800	0.0073
39		${Hpcoh}_{HRV}^{EMGi}$(0.4 Hz)	0.48 ± 0.0100	0.25 ± 0.0300	0.0044
40		${Dscoh}_{HRV}^{EMGi}$	0.93 ± 0.0390	0.47 ± 0.0350	0.0020

**Table 8 sensors-25-06000-t008:** Results of time-varying signal processing.

No.	Parameter	Successful Group (Mean ± SD)	Failure Group (Mean ± SD)	*p*-Value
41	${SampEn}^{EMGe}$	1.8049 ± 0.3369	1.5611 ± 0.3493	0.0366
42	${ApEn}^{EMGe}$	1.9057 ± 0.2892	1.6335 ± 0.3408	0.0335
43	${SampEn}^{EMGi}$	1.5415 ± 0.3088	1.5197 ± 0.3808	0.0406

SD = standard deviation. The number in the first column corresponds to the label for the 43 selected parameters, which were significant and correlated, as presented in Table 5, Table 6 and Table 7.

**Table 9 sensors-25-06000-t009:** Low correlations between significant parameters identified by the Spearman rank correlation test.

No.	1	4	5	10	41	42	43	17	29	30	31	32	33	39	40
1		−0.1	−0.3	0.0	−0.1	0.1	0.0	−0.3	−0.1	−0.1	−0.2	−0.5	−0.5	−0.2	0.1
4			0.2	0.0	0.1	0.0	0.1	0.0	0.3	0.0	−0.2	−0.1	0.1	−0.2	0.0
5				0.0	0.1	−0.1	0.0	0.2	0.1	−0.1	0.0	0.3	0.1	−0.1	−0.1
10					−0.1	−0.2	−0.1	0.0	−0.1	0.1	−0.1	0.1	−0.1	0.1	0.2
41						0.5	0.0	−0.1	0.1	0.2	−0.2	0.0	−0.2	−0.2	−0.1
42							0.0	0.1	0.1	0.0	0.0	−0.3	0.0	0.0	0.0
43								0.0	0.3	0.1	0.0	−0.2	0.1	−0.4	−0.1
17									−0.1	0.0	0.2	0.0	0.3	−0.1	0.0
29										0.0	0.0	−0.1	0.0	−0.2	0.0
30											0.2	−0.3	0.0	−0.3	0.2
31												0.0	0.4	−0.1	0.2
32													−0.1	0.4	0.0
33														0.1	0.1
39															−0.1
40															

**Table 10 sensors-25-06000-t010:** Summarised classification performance of the Naive Bayes classifier. The results averaged over 1000 runs.

Naive Bayes Classifier (Mean ± SD)				
Attribute Set *	Accuracy	Specificity	Sensibility	F Score
Complete	0.481 ± 0.16	0.738 ± 0.35	0.395 ± 0.22	0.514 ± 0.27
Without No. 1	0.678 ± 0.16	0.566 ± 0.28	0.672 ± 0.26	0.615 ± 0.27
Without No. 1-4	0.691 ± 0.16	0.590 ± 0.27	0.718 ± 0.24	0.648 ± 0.25
Without No. 1-4-5	0.722 ± 0.16	0.602 ± 0.26	0.731 ± 0.24	0.660 ± 0.25
Without No. 1-4-5-42	0.773 ± 0.16	0.610 ± 0.27	0.737 ± 0.24	0.668 ± 0.25
Without No. 1-4-5-42-43	0.784 ± 0.16	0.628 ± 0.26	0.728 ± 0.25	0.674 ± 0.26
Without No. 1-4-5-42-43-31	0.798 ± 0.16	0.637 ± 0.26	0.757 ± 0.25	0.692 ± 0.25
Without No. 1-4-5-42-43-30-31	0.830 ± 0.17	0.643 ± 0.25	0.800 ± 0.24	0.713 ± 0.24
Without No. 1-4-5-42-43-30-31-40	0.855 ± 0.12	0.847 ± 0.21	0.834 ± 0.21	0.840 ± 0.21
Without No. 1-4-5-42-43-17-30-31-40	0.835 ± 0.16	0.788 ± 0.26	0.614 ± 0.23	0.841 ± 0.24

* Attribute set refers to the parameters used for each iteration, which are gradually excluded to enhance the classifier’s performance.

**Table 11 sensors-25-06000-t011:** Summarised classification performance of the KNN classifier. The results averaged over 1000 runs.

KNN Classifier (Mean ± SD)				
Attribute Set *	Accuracy	Specificity	Sensibility	F Score
Complete	0.485 ± 0.15	0.157 ± 0.20	0.731 ± 0.22	0.258 ± 0.21
Without No. 40	0.560 ± 0.14	0.335 ± 0.27	0.676 ± 0.21	0.448 ± 0.24
Without No. 43-40	0.641 ± 0.16	0.508 ± 0.27	0.741 ± 0.21	0.603 ± 0.24
Without No. 43-30-40	0.685 ± 0.16	0.520 ± 0.28	0.774 ± 0.19	0.622 ± 0.23
Without No. 43-29-30-40	0.694 ± 0.15	0.465 ± 0.27	0.831 ± 0.17	0.596 ± 0.22
Without No. 5-43-29-30-40	0.675 ± 0.14	0.458 ± 0.27	0.831 ± 0.17	0.591 ± 0.21
Without No. 5-10-43-29-30-40	0.679 ± 0.16	0.487 ± 0.29	0.824 ± 0.17	0.612 ± 0.21
Without No. 5-10-43-17-29-30-40	0.674 ± 0.16	0.495 ± 0.27	0.808 ± 0.21	0.614 ± 0.23

* Attribute set refers to the parameters used for each iteration, which are gradually excluded to enhance the classifier’s performance.

**Table 12 sensors-25-06000-t012:** Summarised classification performance of the SVM classifier. The results averaged over 1000 runs.

SVM Classifier (Mean ± SD)				
Attribute Set *	Accuracy	Specificity	Sensibility	F Score
Complete	0.478 ± 0.14	0.615 ± 0.39	0.375 ± 0.32	0.466 ± 0.35
Without No. 5	0.578 ± 0.16	0.412 ± 0.28	0.703 ± 0.28	0.519 ± 0.28
Without No. 5-39	0.609 ± 0.16	0.397 ± 0.30	0.716 ± 0.27	0.511 ± 0.29
Without No. 5-33-39	0.649 ± 0.16	0.487 ± 0.28	0.626 ± 0.28	0.548 ± 0.28
Without No. 5-31-33-39	0.657 ± 0.17	0.412 ± 0.31	0.736 ± 0.28	0.528 ± 0.29
Without No. 5-42-31-33-39	0.670 ± 0.17	0.458 ± 0.26	0.753 ± 0.26	0.570 ± 0.26
Without No. 5-42-17-31-33-39	0.648 ± 0.16	0.495 ± 0.27	0.723 ± 0.27	0.588 ± 0.27

* Attribute set refers to the parameters used for each iteration, which are gradually excluded to enhance the classifier’s performance.

**Table 13 sensors-25-06000-t013:** Comparison of results on the same data set under different types of classifiers.

Classifier	Accuracy	Specificity	Sensibility	F Score
	(Mean ± SD)			
KNN	0.855 ± 0.12	0.847 ± 0.21	0.834 ± 0.21	0.840 ± 0.21
SVM	0.831 ± 0.15	0.848 ± 0.22	0.818 ± 0.19	0.833 ± 0.20

## Data Availability

The data supporting the findings of this study are available from the corresponding author upon reasonable request. Data sharing is restricted due to privacy and ethical considerations involving patient information.

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
