# Peer review of "Electromyographic Diaphragm and Electrocardiographic Signal Analysis for Weaning Outcome Classification in Mechanically Ventilated Patients"

_sensors, 2025, doi:10.3390/s25196000_

Round 1
Reviewer 1 Report
Comments and Suggestions for Authors
The authors present an analysis aimed at predicting weaning outcomes in mechanically ventilated patients using ECG and EMG signals. While the concept is interesting, several concerns need to be addressed before the manuscript can be considered for publication:
-
-
Please remove the subheadings from the abstract. The abstract should be written as a single, concise paragraph.
-
What is the difference in ECG signals between successful and unsuccessful extubation patients? Both signal features appear similar. Please justify any differences based on characteristic peaks or the presence of abnormal peaks.
-
Why is the EMG amplitude higher in unsuccessful extubation patients compared to successful ones? In addition, please improve the quality and resolution of the figures, as the horizontal and vertical scales are currently unclear.
-
The HRV signal appears highly random and may represent noise. How do the authors validate that the HRV signal contains meaningful information? Please clarify.
-
The authors should compare signals using Signal-to-Noise Ratio (SNR) rather than only amplitude or trend. Please include SNR values for all ECG and EMG signals.
-
Why do the PSD amplitude averages for successful and unsuccessful groups appear almost identical? The interpretation of Fig. 8 is unclear and requires clarification.
-
Author Response
Dear Reviewer,
We sincerely thank you for the time, effort, and valuable insights you have provided in reviewing our manuscript entitled:
"Electromyographic diaphragm and electrocardiographic signal analysis for weaning outcome classification in mechanically ventilated patients."
Please find below our detailed, point-by-point responses to your comments and suggestions:
- Please remove the subheadings from the abstract. The abstract should be written as a single concise paragraph.
Response:
We appreciate this recommendation. The subheadings have been removed, and the abstract is now presented as a single concise paragraph, in accordance with the journal’s guidelines.
- What is the difference in ECG signals between successful and unsuccessful extubation patients? Both signal features appear similar. Please justify any differences based on characteristic peaks or the presence of abnormal peaks.
Response:
We have included an explanatory paragraph describing the visual features observed in Figure 2, which shows ECG examples from a successful and an unsuccessful extubation case. Although the ECG morphology may appear visually similar, our analysis focused on ECG-derived variables, such as the ECG-Derived Respiration (EDR) signal and Heart Rate Variability (HRV). Patients who underwent successful extubation exhibited higher EDR amplitude and greater modulation in the high-frequency (HF) band, indicating improved cardiorespiratory coupling. These differences were statistically significant (p < 0.05) and have been clarified in the Results section and further elaborated in the Discussion.
- Why is the EMG amplitude higher in unsuccessful extubation patients compared to successful ones? In addition, please improve the quality and resolution of the figures, as the horizontal and vertical scales are currently unclear.
Response:
An explanatory paragraph has been added to clarify the findings in Figure 3. The higher EMG amplitude observed in the unsuccessful extubation group reflects increased respiratory effort, possibly due to muscle fatigue or inefficient ventilation. This observation is consistent with previous findings that associate increased diaphragmatic activity with extubation failure. This point has been clarified and supported with relevant references in the Discussion section. Regarding the figures, we have updated Figures 2, 3, 4, 6, 7, 8, 9, and 11 with improved resolution, and clearly visible horizontal and vertical scales have been edited to enhance interpretability.
- The HRV signal appears highly random and may represent noise. How do the authors validate that the HRV signal contains meaningful information? Please clarify.
Response:
To address this concern, we calculated the Signal-to-Noise Ratio (SNR) of the HRV signals, obtaining a mean value of 18.39 ± 3.12 dB. In addition, we replaced the subject originally shown in Figure 6 with another subject presenting a higher SNR also positive but more representative of the majority of participants in the study. The HRV signal was preprocessed using artifact removal techniques and outlier correction via cubic interpolation, as detailed in the Methods section. Furthermore, we applied both linear (Power Spectral Density, coherence) and nonlinear analyses (Approximate Entropy [ApEn], Sample Entropy [SampEn]). The results demonstrated statistically significant differences between groups (p < 0.05), supporting the physiological validity of the HRV signal. This explanation has been reinforced and expanded upon in the revised manuscript.
- The authors should compare signals using Signal-to-Noise Ratio (SNR) rather than only amplitude or trend. Please include SNR values for all ECG and EMG signals.
Response:
We thank the reviewer for this valuable suggestion. We have calculated SNR values for both the ECG and EMG signals, as well as for their derived signals. The average SNR values have been included in the Results section, and the corresponding methodology has been described in the Methods section of the revised manuscript.
- Why do the PSD amplitude averages for successful and unsuccessful groups appear almost identical? The interpretation of Fig. 8 is unclear and requires clarification.
Response:
We appreciate this important observation. The cardiorespiratory system operates within relatively narrow physiological ranges, and subtle deviations can reflect dysfunction or insufficient readiness for extubation. While the average differences in PSD amplitude may appear small, these nuanced variations—when integrated across multiple features—can be effectively captured by classification algorithms such as those used in this study (Naive Bayes, KNN, SVM). As demonstrated, the Naive Bayes classifier achieved 85% accuracy, highlighting the clinical utility of combining such features. We have enhanced the visual quality of Figure 8 and expanded its interpretation in the Results section to provide a clearer explanation of these findings.
Reviewer 2 Report
Comments and Suggestions for Authors
The article addresses a critical clinical issue with an innovative technical approach based on biomedical signals and machine learning techniques. However, to strengthen its scientific and clinical impact, the following is recommended:
1. Expand the state of the art with more recent studies on the use of sEMG and ECG in clinical prediction, including deep learning approaches such as CNN-LSTM or Transformers, which have already been applied to biomedical time series with better generalisation rates.
2. Clarify the clinical justification for certain spectral and non-linear parameters, as well as discuss why techniques such as entropy or spectral coherence would be superior to other metrics.
3. Evaluate the external validity of the model. The absence of an independent cross-validation set or comparison with clinical predictions limits the scope of the work.
4. Review the technical language of the manuscript in English and avoid unnecessary repetition of technical descriptions.
5. Better contextualise the clinical implications, especially when making claims about potential use in ICUs. Be more cautious and show the need for validation in multicentre cohorts.

There are sentences with confusing grammatical structures and incorrect verb tenses. I recommend having it reviewed by a native speaker or a professional proofreader.
Author Response
Dear Reviewer,
We sincerely thank you for the time, effort, and valuable insights you have provided in reviewing our manuscript entitled:
"Electromyographic diaphragm and electrocardiographic signal analysis for weaning outcome classification in mechanically ventilated patients."
Please find below our detailed, point-by-point responses to your comments and suggestions:
- Expand the state of the art with more recent studies on the use of sEMG and ECG in clinical prediction, including deep learning approaches such as CNN-LSTM or Transformers, which have already been applied to biomedical time series with better generalisation rates.
Response:
We appreciate this valuable suggestion. We have expanded the Introduction section to include recent references related to the use of Convolutional Neural Networks (CNNs), Recurrent Neural Networks (RNNs), hybrid CNN-LSTM models, and Transformer architectures in the classification of biomedical signals. Although our study focused on traditional classifiers to enhance clinical interpretability and to facilitate the selection of physiologically meaningful features, we acknowledge the strong potential of deep learning models in this field and consider them promising avenues for future research.
- Clarify the clinical justification for certain spectral and non-linear parameters, as well as discuss why techniques such as entropy or spectral coherence would be superior to other metrics.
Response:
Thank you for this insightful suggestion. We have expanded the Introduction to provide a clearer clinical rationale for the use of parameters such as entropy (ApEn, SampEn) and spectral coherence (MSC). These metrics are particularly relevant in the context of weaning from mechanical ventilation, as they capture the complexity and synchronization of the cardiorespiratory and neuromuscular systems. Unlike simpler metrics such as mean amplitude or RMS, entropy measures the irregularity and unpredictability of physiological signals, which can indicate diaphragmatic fatigue or autonomic dysfunction. Spectral coherence, on the other hand, quantifies the functional coupling between physiological signals such as those from the heart and diaphragm and provides insights into their coordinated activity, which is critical for evaluating patient readiness for extubation.
- Evaluate the external validity of the model. The absence of an independent cross-validation set or comparison with clinical predictions limits the scope of the work.
Response:
We acknowledge this limitation. The external validation of the proposed model is beyond the scope of the present study. Nevertheless, we have explicitly addressed this limitation in the Conclusions section, where we recommend that future studies incorporate independent validation datasets and comparisons with clinical predictions in order to improve the generalizability of the findings.
- Review the technical language of the manuscript in English and avoid unnecessary repetitions of technical descriptions.
Response:
We have thoroughly revised the technical language throughout the manuscript. Redundant technical descriptions have been removed, and the overall writing has been improved to enhance clarity, coherence, and readability in English.
- Better contextualise the clinical implications, especially when making claims about potential use in ICUs. Be more cautious and show the need for validation in multicentre cohorts.
Response:
We appreciate this critical observation. We have moderated the tone of the clinical claims. While the results are promising, we conclude by emphasizing the need for further validation in larger and multicentre cohorts before considering implementation in Intensive Care Units. We also highlight the importance of integrating these findings with expert clinical judgment and the individual patient context.
Reviewer 3 Report
Comments and Suggestions for Authors
The manuscript presents a clinically relevant study on predicting weaning outcomes in mechanically ventilated patients using combined EMG and ECG signal analysis. There are some comments:
- In P. 15, it said “ables 5 and 6 present the mean values, coefficient of variation…”, please check it.
- In Fig. 11, it shows the Comparison of Classifiers. Why were the three algorithms not using the same parameters and not having the same number of parameters? In the description, it said “KNN classifier achieves its highest performance by excluding 7 parameters”, please check it.
- In the results presented in Table 3, why is there no result for KNN?
- 4. In the references, the No.10 and No. 13 are the same.
Author Response
Dear Reviewer,
We sincerely thank you for the time, effort, and valuable insights you have provided in reviewing our manuscript entitled:
"Electromyographic diaphragm and electrocardiographic signal analysis for weaning outcome classification in mechanically ventilated patients."
Please find below our detailed, point-by-point responses to your comments and suggestions:
- In P. 15, it said “Tables 5 and 6 present the mean values, coefficient of variation…”, please check it.
Response:
We thank the reviewer for pointing out this typographical error. It has been corrected in the revised manuscript and now reads accurately as: “Tables 6 and 7 present...”.
- In Fig. 11, it shows the Comparison of Classifiers. Why were the three algorithms not using the same parameters and not having the same number of parameters? In the description, it said “KNN classifier achieves its highest performance by excluding 7 parameters”, please check it.
Response:
Thank you for this observation. Each classifier was independently optimized using a sequential feature selection strategy to maximize its individual performance. As a result, the number and combination of input parameters differed among classifiers. A key aspect of our analysis is that each algorithm achieved its best performance with a distinct subset of features, which highlights differences in how they interpret and prioritize the input data. This iterative parameter elimination process revealed valuable insights into the behavior of each classifier. Additionally, we corrected the description in the manuscript: the KNN classifier achieved its highest performance by excluding six parameters, not seven.
- In the results presented in Table 13, why is there no result for KNN?
Response:
We regret the omission. The performance metrics for the KNN classifier were detailed in Table 11, but Table 3 was misnumbered in an earlier draft. This has been corrected, and we ensured all classifiers (Naïve Bayes, SVM, KNN) are consistently reported across all tables and figures. Table 13 and Figure 12 were elaborated to show the performance of KNN and SVM classifiers with the obtained variables from Naive Bayes. Due to a misspelling, the idea was not clear in the previous draft. This is now corrected.
- In the references, the No.10 and No.13 are the same.
Response:
Thank you for spotting this duplication. We have removed the redundant reference and updated the numbering of all subsequent references accordingly in both the reference list and in text citations.
Round 2
Reviewer 1 Report
Comments and Suggestions for Authors
The authors has revised the manuscript well. Thus, the manuscript can be accepted at its current form